Influence of whole-wheat consumption on fecal microbial community structure of obese diabetic mice

http://orcid.org/0000-0001-6047-4361 Garcia-Mazcorro Jose F. 1 2 josegarcia_mex@hotmail.com
Ivanov Ivan 3
Mills David A. 4
Noratto Giuliana 5 # gnoratto@tamu.edu
1 Faculty of Veterinary Medicine, Universidad Autónoma de Nuevo León , General Escobedo, Nuevo Leon , Mexico
2 Research Group Medical Eco-Biology, Universidad Autónoma de Nuevo León , General Escobedo, Nuevo Leon , Mexico
3 Veterinary Physiology and Pharmacology, Texas A&M University , College Station, Texas , United States
4 Department of Food Science and Technology, University of California, Davis , Davis, California , United States
5 School of Food Science, Washington State University , Pullman, Washington , United States
# Current Address: Nutrition and Food Science, Texas A&M University , College Station, Texas , United States
Howe Adina
Electronic publication date: 2016 Feb 15
Publication date: 2016
Volume: 4
Electronic Location ID: e1702
Received 2015 Oct 28; Accepted 2016 Jan 27
Copyright: © 2016 Garcia-Mazcorro et al.
Copyright year: 2016
Copyright holder: Garcia-Mazcorro et al.
License: This is an open access article distributed under the terms of the Creative Commons Attribution License, which permits unrestricted use, distribution, reproduction and adaptation in any medium and for any purpose provided that it is properly attributed. For attribution, the original author(s), title, publication source (PeerJ) and either DOI or URL of the article must be cited.
License URL: https://creativecommons.org/licenses/by/4.0/

Keywords: Fecal microbiota, High-Throughput sequencing, Metabolic pathways, Obesity, Whole-wheat

Funding: Washington Gran Commission 3057–4668 This work was supported by the Washington Gran Commission (Grant number: 3057–4668). JFGM received financial support from CONACYT (Mexico) through the National System of Researchers (SNI, for initials in Spanish) program and PRODEP (Mexico). DAM received support from the Peter J. Shields Endowed Chair. The funders had no role in study design, data collection and analysis, decision to publish, or preparation of the manuscript.

==============================
The digestive tract of mammals and other animals is colonized by trillions of metabolically-active microorganisms. Changes in the gut microbiota have been associated with obesity in both humans and laboratory animals. Dietary modifications can often modulate the obese gut microbial ecosystem towards a more healthy state. This phenomenon should preferably be studied using dietary ingredients that are relevant to human nutrition. This study was designed to evaluate the influence of whole-wheat, a food ingredient with several beneficial properties, on gut microorganisms of obese diabetic mice. Diabetic (db/db) mice were fed standard (obese-control) or whole-wheat isocaloric diets (WW group) for eight weeks; non-obese mice were used as control (lean-control). High-throughput sequencing using the MiSeq platform coupled with freely-available computational tools and quantitative real-time PCR were used to analyze fecal bacterial 16S rRNA gene sequences. Short-chain fatty acids were measured in caecal contents using quantitative high-performance liquid chromatography photo-diode array analysis. Results showed no statistical difference in final body weights between the obese-control and the WW group. The bacterial richness (number of Operational Taxonomic Units) did not differ among the treatment groups. The abundance of Ruminococcaceae, a family containing several butyrate-producing bacteria, was found to be higher in obese (median: 6.9%) and WW-supplemented mice (5.6%) compared to lean (2.7%, p = 0.02, Kruskal-Wallis test). Caecal concentrations of butyrate were higher in obese (average: 2.91 mmol/mg of feces) but especially in WW-supplemented mice (4.27 mmol/mg) compared to lean controls (0.97 mmol/mg), while caecal succinic acid was lower in the WW group compared to obese but especially to the lean group. WW consumption was associated with ∼3 times higher abundances of Lactobacillus spp. compared to both obese and lean control mice. Analysis of weighted UniFrac distances revealed a distinctive clustering of lean microbial communities separately from both obese and WW-supplemented mice (p = 0.001, ANOSIM test). Predictive metagenome analysis revealed significant differences in several metabolic features of the microbiota among the treatment groups, including carbohydrate, amino acids and vitamin metabolism (p < 0.01, Kruskal-Wallis test). However, obese and WW groups tended to share more similar abundances of gene families compared to lean mice. Using an in vivo model of obesity and diabetes, this study suggests that daily WW supplementation for eight weeks may not be enough to influence body weight or to output a lean-like microbiome, both taxonomically and metabolically. However, WW-supplementation was associated with several statistically significant differences in the gut microbiome compared to obese controls that deserve further investigation.

Introduction

Obesity is an epidemic with catastrophic consequences for the health of millions of people around the globe. Different strategies can help reduce body weight including changes in exercise and dietary habits, yet many patients genuinely struggle to successfully decrease their body weight due to multiple interrelated factors (Gupta, 2014).

The mammalian digestive tract is a complex organ that has been constantly co-evolving with trillions of microorganisms (the gut microbiota) to combat pathogens and maximize food digestion for at least 600 million years. Despite its general resilience, the gut microbiota is still susceptible to changes in dietary and other life habits, some of which can lead to imbalances and consequently to disease (Lozupone et al., 2012). For instance, substantial evidence has been published showing an association between obesity and changes in gut microbial populations and its metabolism of dietary and endogenous compounds (Delzenne & Cani, 2011). Interestingly, the changes in gut microbial communities between lean and obese individuals are not irreversible (Turnbaugh et al., 2008) with diet being the most practical alternative to reestablish microbial equilibrium within the gut. Understanding changes in gut microorganisms in response to dietary modifications is essential to develop effective dietary strategies to help obese patients.

Growing evidence shows that the consumption of specific dietary ingredients or supplements such as probiotics, prebiotics, polyphenols, as well as whole-grains has the potential of modifying gut health parameters in obese individuals, both in humans and animal models (Katcher et al., 2008; Noratto et al., 2014; Petschow et al., 2013; Vitaglione et al., 2015). Whole-Wheat (WW) is often recommended by medical nutritionists as part of a healthy diet for both overweighed and lean individuals. While several investigations have previously addressed the nutritional benefits of consuming WW (Stevenson et al., 2012), very few studies have researched the potential of either WW or its individual nutrients to alter the gut microbiota of lean or obese individuals (Neyrinck et al., 2011) or as part of dietary management to treat obesity. One study investigated the effect of replacing refined wheat with whole-grain wheat for 12 weeks on body weight and fat mass in overweighed women (Kristensen et al., 2011). This short-period of 12 weeks was enough to significantly reduce percentage fat mass but no body weights. Here we show that an 8-week consumption period of an isocaloric WW diet did not significantly change body weights in obese-diabetic mice. Overall, obese mice under WW-supplemented diet showed similarities to obese controls with regards to gut microbial composition and predicted metabolic profile. The effect of WW was mostly observed on caecal concentrations of butyrate and succinate and a few bacterial groups such as Lactobacillus. The results may have implications in clinical dietary management of obesity using WW.

Methods

Study design

The Institutional Animal Care Use Committee from Washington State University approved all experimental procedures (animal protocol approval number: 04436-001). Two strains of male mice were used in this study, BKS.Cg- + Leprdb/+Leprdb/OlaHsd obese diabetic (db/db), and lean BKS.Cg-Dock7m +/+ Leprdb/OlaHsd (Harlan Laboratories, Kent, WA). Animals were purchased at 5–6 weeks of age and maintained in ventilated rack system with food and water provided ad libitum throughout the study. We received 11 mice for the lean group and 10 mice from all other groups. After 7 days of acclimatization, obese mice were randomly divided into two groups (n = 10 each) namely obese (AIN-93 G Purified Rodent Diet) and WW (whole-wheat supplemented diet). The wild type mice group (n = 11) was named lean (AIN-93 Diet). Diets were made by Dyets Inc. (Bethlehem, PA) (Table 1). Four or five mice per cage were housed in an environment-controlled room (23 °C, 12 hours dark-light cycle). All mice were visually inspected every day and body weight was recorded from all animals once a week.

Table 1 Formulation of experimental diets (g/100 g).

Ingredients	Lean and obese diet	Wheat diet	
Casein, high nitrogen	20	0.0	
L-Cysteine	0.3	0.3	
Whole-wheat meal	0.0	87.94	
Soybean oil	7.0	7.0	
Sucrose	10	0.0	
Cornstarch	39.74	0.0	
Dyetrose	13.2	0.0	
t-Butylhydroquinone	0.0014	0.0014	
Cellulose	5	0.0	
Mineral mix #210025	3.5	3.5	
Vitamin mix #310025	1.0	1.0	
Choline bitartrate	0.25	0.25	
Kcal/100 g	376.00	387.76	

Fecal collection and DNA extraction

Fresh distal colon contents (see qPCR analysis below) and fecal samples were obtained from all mice at the end of the study (8 weeks) and stored at −80 °C prior to DNA and 16S rRNA gene profiling analysis. Total DNA was extracted from at least two different fecal pellets weighting approximately 200 mg. Following bead-beating, the QIAamp DNA Stool Mini Kit (Qiagen Inc., Valencia, CA, USA) was used for DNA extraction following the manufacturer’s instructions. DNA concentration and purity was determined using a NanoDrop Spectrophotometer (Thermo Scientific, Wilmington, DE, USA) and diluted to a working concentration of 5 ng/μL.

High-throughput sequencing of 16S rRNA genes

Amplification and sequencing were performed as described elsewhere (Bokulich et al., 2014). Briefly, the V4 semi-conserved region of bacterial 16S rRNA genes was amplified using primers F515 (5′-GTGCCAGCMGCCGCGGTAA-3′) and R806 (5′-GGACTACHVGGGTWTCTAAT-3′), with the forward primer modified to contain a unique 8-nt barcode and a 2-nt linker sequence at the 5′ terminus. Amplicons were combined into one pooled sample and submitted to the University of California Davis Genome Center DNA Technologies Core for Illumina paired-end library preparation, cluster generation, and 250-bp paired-end sequencing on an Illumina MiSeq instrument in one runs. For data analysis, raw Illumina fastq files were demultiplexed, quality filtered, and analyzed using the freely available Quantitative Insights into Microbial Ecology (QIIME) Virtual Box v.1.8.0 (Caporaso et al., 2010). Operational Taxonomic Units (OTUs) were assigned using two different approaches: first, using UCLUST v.1.2.22 (Edgar, 2010) as implemented in QIIME using the open-reference clustering algorithm described in (Rideout et al., 2014) for alpha and beta diversity analyses; and second, using the pick_closed_reference_otus.py QIIME script for further analysis using PICRUSt (see Predicted metabolic profiles below). The Greengenes 13_5 97% OTU representative 16S rRNA gene sequences was used as the reference sequence collection (DeSantis et al., 2006). Alfa and beta diversity analyses were performed using 3000 random sequences per sample (lowest number of sequences in a sample after demultiplexing, filtering and OTU picking). Raw sequences were uploaded into the Sequence Read Archive at NCBI (accession number: PRJNA281761). The trim.seqs command in MOTHUR (Schloss et al., 2009) was used for splitting original fastq files per sample for uploading to SRA.

Predicted metabolic profiles

OTUs from the closed_reference script were normalized and used to predict metagenome functional content using the online Galaxy version of Phylogenetic Investigation of Communities by Reconstruction of Unobserved States (PICRUSt) (Langille et al., 2013). PICRUSt uses existing annotations of gene content as well as 16S copy numbers from reference microbial genomes in the IMG database (Markowitz et al., 2012) and a functional classification scheme to catalogue the predicted metagenome content. The current galaxy version supports three types of functional predictions; this current study used the popular KEGG Orthologs (Kanehisa et al., 2012).

Quantitative real-time PCR (qPCR) analysis

DNA was extracted from distal colon content using the ZR Fecal DNA MiniPrep™ kit following the manufacturer’s protocol (Zymo Research, Irvine, CA, USA). qPCR was used to detect specific bacterial groups as described elsewhere (Noratto et al., 2014). Table 2 shows the primers sequences used for all qPCR analyses.

Table 2 Oligonucleotides used in this study for qPCR analyses.

qPCR primers	Sequence (5′-3′)	Target	Reference	
HDA1	ACTCCTACGGGAGGCAGCAGT	All bacteria (V2–V3 regions, position 339–539 in the E. coli 16S gene)	Walter et al. (2000)	
HDA2	GTATTACCGCGGCTGCTGGCAC			
Bact834F	GGARCATGTGGTTTAATTCGATGAT	Bacteroidetes (Phylum)	Guo et al. (2008)	
Bact1060R	AGCTGACGACAACCATGCAG			
928F-Firm	TGAAACTYAAAGGAATTGACG	Firmicutes (Phylum)	Bacchetti et al. (2011)	
1040firmR	ACCATGCACCACCTGTC			
BifF	GCGTGCTTAACACATGCAAGTC	Bifidobacterium (genus)	Penders et al. (2005)	
BifR	CACCCGTTTCCAGGAGCTATT			
E. coli F	CATGCCGCGTGTATGAAGAA	E. coli	Huijsdens et al. (2002)	
E. coli R	CGGGTAACGTCAATGAGCAAA			
TuriciF	CAGACGGGGACAACGATTGGA	Turibacter (genus)	Suchodolski et al. (2012)	
TuriciR	TACGCATCGTCGCCTTGGTA			
RumiF	ACTGAGAGGTTGAACGGCCA	Ruminococcaceae (family)	Garcia-Mazcorro et al. (2012)	
RumiR	CCTTTACACCCAGTAAWTCCGGA			
FaecaliF	GAAGGCGGCCTACTGGGCAC	Faecalibacterium (genus)	Garcia-Mazcorro et al. (2012)	
FaecaliR	GTGCAGGCGAGTTGCAGCCT			
Eco1457-F	CATTGACGTTACCCGCAGAAGAAGC	Enterobacteriaceae (family)	Bartosch et al. (2004)	
Eco1652-R	CTCTACGAGACTCAAGCTTGC			
V1F	CAGCACGTGAAGGTGGGGAC	Akkermansia muciniphila	Collado et al. (2007)	
V1R	CCTTGCGGTTGGCTTCAGAT			
PrevF	CACCAAGGCGACGATCA	Prevotella (genus)	Larsen et al. (2010)	
PrevR	GGATAACGCCYGGACCT			
Bfr-F	CTGAACCAGCCAAGTAGCG	Bacteroides fragilis	Liu et al. (2003)	
Bfr-R	CCGCAAACTTTCACAACTGACTTA			

Measurement of Short Chain Fatty Acids (SCFAs) in caecal contents

SCFAs were quantified as reported elsewhere (Campos et al., 2012). Briefly, samples were analyzed by an HPLC-PDA system using an Aminex HPX-87H strong cation-exchange resin column (300 × 7.8 mm) and fitted with an ion exchange microguard refill cartridge (Bio-Rad, Hercules, CA, USA). The HPLC-PDA system consisted of a Water 2695 Separation Module (Waters, Milford, MA), which was equipped with a Water 2996 Photodiode Array detector (PDA). Samples (20 μL) were eluted isocratically with 5 mM sulfuric acid at 0.6 mL/min, and the column temperature was held at 50 °C. Sodium butyrate, acetic acid, oxalic acid, and succinic acid were identified and quantified by comparing retention time and UV-Visible spectral data to standards.

Statistical analysis

ANOVA and the non-parametric alternative Kruskal-Wallis test were used to analyze final body weights and SCFAs concentrations, respectively. Multiple-comparisons were performed using Tukey and Mann-Whitney tests. The Bonferroni (for Mann-Whitney tests) and False Discovery Rate (for Tukey’s tests) corrections were used to adjust for multiple comparisons. Analysis of Similarities (ANOSIM) was used to test for clustering of microbial communities using weighted and unweighted UniFrac distance matrices. QIIME v.1.8.0, R v.3.0.3 (R Core Team, 2014), PAST (Hammer, Harper & Ryan, 2001) and Excel were used for statistics and graphics. The Linear Discriminant Analysis (LDA) Effect Size (LEfSe) method was used to assess differences in microbial communities using a LDA score threshold of 3 (Segata et al., 2011). STAMP (Parks & Beiko, 2010) was used to visualize and analyze the PICRUSt data with ANOVA and False Discovery Rate. Unless otherwise noted, an alpha of 0.05 was considered to reject null hypotheses.

Results

One mouse in the obese group died for reasons unrelated to this study. At the end of the study, there was a significant (p < 0.01, ANOVA) difference in body weight between the lean (average: 30.6 ± 2.2 g) and both the obese (46.1 ± 2.8 g) and WW groups (45.3 ± 5.8 g). WW consumption was not associated with a lower body weight compared to obese control group (p = 0.96, Tukey’s test).

Fecal microbiota composition

A total of 8686 different OTUs were detected using the open reference algorithm described by Rideout et al. (2014). On the other hand, the closed_reference method used to generate data for PICRUSt (see PICRUSt below) only yielded 1302 OTUs. Fecal microbial composition of all mice was mostly comprised by Firmicutes (average: 58.7% across all samples) and Bacteroidetes (average: 32.8%) (Fig. 1). Other less abundant Phyla were Actinobacteria (∼4%), Proteobacteria (∼3%), Verrucomicrobia (∼0.8%) and others (Fig. 1). There was no statistical difference in relative abundance of the two most abundant phyla (Firmicutes and Bacteroidetes), partly because of the high variability among individual mice. The ratio Bacteroidetes/Firmicutes was lower in the lean (median: 38.7%) compared to the obese group (median: 85.3%) and the WW group (median: 75.4%) but this difference did not reach significance (p = 0.12, Kruskal-Wallis). Several statistical differences were found in low abundant phyla. Actinobacteria and Verrucomicrobia were higher in lean, Cyanobacteria, TM7 and Tenericutes were higher in the WW group, and Deferribacteres was higher in obese-control (p < 0.01, Kruskal-Wallis, Fig. 1).

Figure 1 Box plots.

Composition of fecal microbiota at the phylum level in the Lean (n = 11), obese control (Obese, n = 9) and whole-wheat (Wheat, n = 10) group. Boxes represent the 25–75 quartiles, the median is shown with a vertical line inside the box. Values outside 1.5 times the box height are shown as circles; values outside 3 times the box height are shown as stars. The differences in relative abundance of Firmicutes, Bacteroidetes and Proteobacteria did not reach statistical significance (see main text for details).

LEfSe showed statistical significant differences for several microbial groups at lower taxonomic levels (Fig. 2). Among the bacterial groups that showed differences indicating an effect of WW-supplementation include the genus Lactobacillus, the class Gammaproteobacteria and the controversial S24-7 family http://groups.google.com/forum/#!topic/qiime-forum/Ds75aZoVrFY (Fig. 2). Other differences in bacterial abundances suggested that WW-supplementation did not generate a lean-like microbiome. For example, the genera Bifidobacterium, Allobaculum and Akkermansia were higher in lean compared to both obese and WW group (Fig. 2). Also, the family Ruminococcacea was more similar between obese (median: 6.9%) and WW-supplemented mice (5.6%) compared to lean (2.7%). Despite these differences, overall the fecal microbial composition of obese-control and WW mice was more similar to each other compared to lean although WW-supplementation yielded a unique pattern of bacterial abundances that did not necessarily cluster together with all obese samples (Fig. 3).

Figure 2 Bar charts of most significant results using the LDA Effect Size method (LEfSe).

LefSe identifies those bacterial groups that showed statistical significance effect size and associate them with the class (in this study treatment group) with the highest median. (A) lean; (B) obese-control; (C) WW-supplemented obese mice. Dotted lines represent medians; straight lines represent averages.

Figure 3 Heatmap.

Heatmap of relative abundance of the most abundant bacterial taxa at the family level (x axis, ordered by abundance) in Lean (n = 11), Obese (n = 9) and Wheat (n = 10) group. This figure shows that lean subjects clustered separately from obese and WW subjects. Clustering was performed using Bray-Curtis distances in R v.3.2.2.

Alpha diversity

There was no significant difference in number of OTUs and Chao1 diversity index. Interestingly, samples from the obese group showed a more disperse distribution of OTUs (Fig. 4). Rarefied plots of number of OTUs showed that more than the 3000 sequences per sample used in this study are needed to fully describe the fecal microbiota of all mice.

Figure 4 Dispersion in number of OTUs detected.

(A) shows the relationship between the numbers of OTUs and the proportion of samples containing those OTUs for each treatment group. These plots show that more obese control samples contained higher numbers of OTUs compared to whole-wheat.Lines were used to illustrate 50% of the samples (vertical line) and 1000 OTUs (horizontal line). (B) shows boxplots to illustrate the distributions of the number of OTUs for each treatment group (boxes represent the 25–75 quartiles, the median is shown with a horizontal line inside the box).

Beta-diversity

Principal Coordinate Analysis (PCoA) of weighted and unweighted UniFrac metrics showed different clustering of microbial communities. Weighted (which takes phylogenetic information as well as sequence abundance into account) metrics clearly showed a different microbial structure in lean individuals compared to obese and WW groups (ANOSIM, p = 0.001) (Fig. 5). This was expected based on the clustering of lean subjects using relative abundance of sequence reads (Fig. 3). On the other hand, the qualitative (does not take sequence abundance into account) unweighted UniFrac analysis shows that the microbiota of the WW group clustered separately from the lean and obese groups (ANOSIM, p = 0.001) (Fig. 5).

Figure 5 Principal Coordinates Analysis plots.

Principal Coordinates Analysis (PCoA) plots of weighted (A) and unweighted (B) UniFrac distance metrics. Please note that each plot gives contrasting results with regards to the clustering of samples.

Predicted metabolic profile

The taxa predicted by 16S RNA marker gene sequencing was used to predict the functional profile of the fecal microbiome in all three experimental groups. Using a p < 0.01 for ANOVA tests in STAMP, several statistical differences were found (Table 3). Overall, obese and WW groups tended to share more similar abundances of gene families compared to lean mice, an observation that supports the differences in bacterial abundances.

Table 3 Metabolic features in Lean (n = 11), Obese (n = 9) and whole-wheat supplemented (n = 10) mice.

KEGG gene categories	Treatment Groups	p value	
Lean	Obese	Whole-wheat	
Level_1	Level_2	Level_3	Mean ± st. dev.	Mean ± st. dev.	Mean ± st. dev.	
Cellular processes	Transport and catabolism	Peroxisome	0.15 ± 0.03a	0.18 ± 0.04a,b	0.23 ± 0.03c	0.001	
Environmental information processing	Signal transduction	Phosphatidylinositol signaling system	0.11 ± 0.01a	0.08 ± 0.01b	0.10 ± 0.01a,c	0.002	
	Signaling molecules and interaction	Ion channels	0.05 ± 0.01a	0.02 ± 0.01b	0.02 ± 0.01b,c	<0.001	
Genetic information processing	Replication and repair	Base excision repair	0.50 ± 0.06a	0.41 ± 0.03b	0.41 ± 0.04b,c	0.001	
Human diseases	Infectious diseases	Tuberculosis	0.18 ± 0.02a	0.13 ± 0.01b	0.13 ± 0.02b,c	0.001	
	Cancers	Pathways in cancer	0.07 ± 0.01a	0.04 ± 0.00b	0.05 ± 0.00b,c	<0.001	
	Neurodegenerative diseases	Amyotrophic Lateral Sclerosis (ALS)	0.02 ± 0.01a	0.03 ± 0.01b	0.04 ± 0.01b,c	<0.001	
	Cancers	Renal cell carcinoma	0.03 ± 0.01a	0.01 ± 0.00b	0.01 ± 0.01b,c	<0.001	
Metabolism	Carbohydrate metabolism	Fructose and mannose metabolism	1.16 ± 0.19a	0.94 ± 0.08b	0.86 ± 0.12b,c	0.005	
		Pentose phosphate pathway	0.92 ± 0.05a	0.86 ± 0.07a,b	0.78 ± 0.06c	0.002	
	Metabolism of cofactors and vitamins	Porphyrin and chlorophyll metabolism	0.55 ± 0.15a	0.85 ± 0.13b	0.66 ± 0.14a,c	0.009	
		Vitamin B6 metabolism	0.17 ± 0.02a	0.20 ± 0.02b	0.22 ± 0.01b,c	<0.001	
	Metabolism of other amino acids	Selenocompound metabolism	0.40 ± 0.02a	0.37 ± 0.01b	0.35 ± 0.01b,c	<0.001	
		Cyanoamino acid metabolism	0.23 ± 0.04a	0.33 ± 0.02b	0.30 ± 0.06b,c	0.002	
		Beta-alanine metabolism	0.18 ± 0.04a	0.22 ± 0.04	0.28 ± 0.06b	0.005	
	Carbohydrate metabolism	C5-branched dibasic acid metabolism	0.23 ± 0.06a	0.32 ± 0.02b	0.31 ± 0.02b,c	0.003	
	Biosynthesis of other secondary metabolites	Phenylpropanoid biosynthesis	0.12 ± 0.03a	0.20 ± 0.02b	0.17 ± 0.05	0.003	
	Metabolism of terpenoids and polyketides	Biosynthesis of ansamycins	0.14 ± 0.04a	0.11 ± 0.02	0.09 ± 0.02b	0.006	
	Biosynthesis of other secondary metabolites	Flavonoid biosynthesis	0.02 ± 0.01a	0.01 ± 0.00b	0.00 ± 0.00b,c	<0.001	
		Stilbenoid, diarylheptanoid and gingerol biosynthesis	0.01 ± 0.01a	0.00 ± 0.00b	0.00 ± 0.00b,c	0.007	
Note:

This table only shows features that reached statistical significant differences (ANOVA, p < 0.1 adjusted for False Discovery Rate in STAMP). Different letters(a,b,c) indicate statistical significant difference (Tukey’s post-hoc tests, p < 0.05).

qPCR assessment of microbiota in distal colon contents

We performed qPCR analysis for bacterial groups of interest to health in distal colon contents. Similarly to the sequencing results from fecal samples, qPCR results revealed several differences in relative abundance for different bacterial groups (Fig. 6).

Figure 6 Boxplots.

Quantitative real-time PCR (qPCR) results for selected bacterial groups. Results are expressed as relative abundance of 16S rRNA amplified DNA (all results were normalized to qPCR data for total bacteria). * p < 0.05 against lean; # p < 0.05 against Whole-Wheat (WW).

SCFAs caecal concentrations

There was a statistically significant difference among the treatment groups for several SCFAs in caecal contents (Table 4). Butyrate concentrations were higher in the WW group compared to both the lean and the obese group (p < 0.001, Kruskal-Wallis). Also, WW consumption was associated with lower succinic acid concentrations (p = 0.009, Kruskal-Wallis).

Table 4 Median (minimum-maximum) for all Short-Chain Fatty Acids (SCFAs).

Results are expressed in mmol/mg of caecal contents.

SCFA	Lean	Obese	Whole-wheat	p value	
Sodium butyrate	0.97 (0.15–2.65)a	2.91 (1.47–4.35)b	4.27 (3.05–6.26)b,c	<0.001	
Acetic acid	10.2 (7.7–26.3)	12.0 (8.3–18.7)	15.4 (10.1–31.9)	0.208	
Oxalic acid	15.14(6.68–18.91)a	14.60 (8.78–28.01)	9.96 (6.76–12.15)b	0.033	
Succinic acid	39.84 (15.29–97.63)a	22.97 (3.86–71.18)a,b	3.12 (0.91–63.36)c	0.009	
Note:

Different letters(a,b,c) denote statistical significance. p values come from the Kruskal-Wallis test and multiple comparisons were performed using the Mann-Whitney test and corrected with the Bonferroni method.

Discussion

Obesity is a worldwide epidemic disease that has been associated with changes in the gut microbiome in many different studies. Consumption of whole grains is often recommended by medical nutritionists as part of a healthy diet. To our knowledge, this is the first study evaluating the in vivo effect of WW consumption on fecal bacterial community structure of obese diabetic mice, adding valuable information to the literature with regard to the use and development of dietary strategies to help obese patients.

Ley et al. (2005) showed that lean mice have more Bacteroidetes and less Firmicutes compared to obese mice, a finding that has been reported by several other research groups. However, it is important to note that these observations were division-wide (in other words, there was no specific subgroup such as families or genera that were present high or low in abundance) and, more importantly, that other researchers have found either no difference in Firmicutes and Bacteroidetes between obese and lean (Duncan et al., 2008) or more Bacteroidetes in obese compared to normal-weight individuals (Zhang et al., 2009). Interestingly, in this study sequencing showed no statistical difference in the abundance of both phyla Firmicutes and Bacteroidetes between lean and obese control; nonetheless, two important aspects must be taken into account. First, obese and WW mice were consistently more like each other compared to lean mice with regard to the abundance of both phyla. Also, contrary to the observations by Ley et al. (2005), lean mice had more Firmicutes and less Bacteroidetes compared to both obese and WW mice, a difference that did not reach statistical significance. qPCR confirmed the sequencing results about the abundance of Firmicutes but not Bacteroidetes, maybe due to the use of fecal (sequencing) or colon (qPCR) contents for bacterial analysis. Regardless, differences in taxa abundance at the phylum level have little relevance when considering all their individual groups within. For instance, many bacterial groups at lower taxonomic levels deserve attention, like the mucin-degrader Akkermansia which has been shown to be inversely correlated with body weight in rodents and humans (Everard et al., 2013). Accordingly, both sequencing and qPCR in this current study showed that obese mice had fewer Akkermansia and WW consumption surprisingly helped to decrease its abundance even further. Here it is important to note that a higher abundance in feces does not necessarily imply a higher abundance in the mucus. WW consumption was also associated with much more Lactobacillus spp., a bacterial genus frequently used in probiotic formulations, and the genus Allobaculum was practically absent in both obese and WW groups while lean individuals were heavily colonized by this group. These changes in bacterial abundances deserve more investigation.

Beta diversity metrics are useful to study similarities of microbiomes, which in turn have critical consequences for understanding health and disease processes. Lozupone et al. (2007) explains that quantitative beta-diversity measures (weighted UniFrac distances) are better for revealing community differences that are due to changes in relative taxon (OTUs) abundance, while qualitative (unweighted) are most informative when communities differ by what can live in them. Most studies report either weighted or unweighted but few report both. In this study, weighted analysis showed a clear separation of lean samples from all samples from the obese and WW groups, suggesting that the numbers of OTUs are an important determinant to separate lean microbiomes from obese individuals with and without WW. In this study, the results of weighted analysis also show that animal genetics was the predominant factor to separate microbial communities. On the other hand, unweighted analysis showed opposite results: lean and obese samples clustered separately from all samples of the WW group, suggesting that WW helped create an environment that favored a phylogenetically different ecosystem. Importantly, the variation explained by the axes is much lower when using unweighted UniFrac. At this point, both methods should be considered for explaining the changes in gut microbiomes in investigations like this study (Lozupone et al., 2007). The discrepancy between the results of weighted and unweighted results suggests that an 8-week period of WW consumption helped change the overall environment in the intestinal lumen, thus modulating what can live and proliferate in it (unweighted results). Thus, the different environment could promote changes in the abundance of specific taxa (weighted results), as shown in this study for several bacterial groups. Given that the assessment of microbial diversity is a major component in microbial ecological studies and closely relates to our understanding of health and health deviations, we expect others to start inspecting and reporting both weighted and unweighted UniFrac distance metrics. The use of both metrics has been shown to be useful in various investigations (Campbell et al., 2015; Igarashi et al., 2014; Wu et al., 2010).

Microbial butyrate is essential for colon health and lower concentrations of this fatty acid are usually considered non-optimal for gut health (Donohoe et al., 2011). Nonetheless, studies have shown that obese individuals actually have higher fecal butyrate and other SCFAs compared to lean individuals (Fernandes et al., 2014), an observation that suggests that both lower and higher butyrate concentrations than normal may be associated with and perhaps aggravate disease. Similarly, obese mice in this current study (with and without WW supplementation) had higher butyrate concentrations in caecal contents compared to lean mice. Butyrate-producers are abundant in the mammalian gut and mainly belong to the family Ruminococcaceae within the Firmicutes (Louis & Flint, 2009). In this study both sequencing and qPCR revealed higher fecal Ruminococcaceae in obese and WW groups compared to lean individuals, thus potentially explaining the higher caecal butyrate concentrations. Another SCFA that deserves attention is succinic acid, which has been shown to increase in rats fed a high-fat diet (Jakobsdottir et al., 2013). In this current study, obese mice had lower concentrations of succinic acid and WW-supplementation seemingly helped to drastically decrease it. Unfortunately, far more attention has been paid to butyrate compared to succinate, propionate and other SCFA (Cheng et al., 2013; Reichardt et al., 2014).

The assessment of microbial metabolic activity in complex ecosystems is hampered in part by the huge number of microorganisms and the cost of sequencing either whole genomes or transcriptomes. PICRUSt allows a prediction of the metabolic profile using taxa predicted by 16S rRNA gene sequencing. PICRUSt is, however, not exempt of pitfalls: it only uses information for well-defined 16S sequences and the presence of a given set of genes does not tell anything about their functional activity depending on the specific environmental conditions. Supported by the similarities in abundance of most bacterial groups between obese-control and WW groups, this study showed that 8-week WW consumption was not enough to make a significant difference in the abundance of bacterial gene families.

Caveats

This study was designed to obtain preliminary information about the influence of WW consumption on gut microbial ecology of obese diabetic mice; therefore, we did not aim to determine the exact compound(s) behind the observed effects. Wheat is a fiber-rich grain and consumption of fiber alone is associated with changes in the gut microbiome and the immune system of the host (Bermudez-Brito et al., 2015). Aside from fiber, WW also contains other bioactive compounds (e.g. polyphenols) that may be responsible for specific effects on host metabolism, physiology and immune system. For instance, it has been recently shown that wheat-derived alkylresorcinols were capable of showing beneficial effects on diet-induced obese mice (Oishi et al., 2015). Interestingly, our group showed that carbohydrate-free polyphenol-rich juice from plum is capable of impeding body weight gain in obese Zucker rats (Noratto et al., 2014), a finding that was not observed with WW consumption in mice in this current study. More research is necessary to investigate the separate effect of the different nutrients in WW.

Summary

In summary, this study suggests that an 8-week consumption of whole-wheat may not be enough to exert an effect on body weight and to output a lean-like microbiome using an in vivo model of obesity and diabetes. However, WW-supplementation was associated with several statistically significant changes compared to obese controls that deserve further investigations. These results may or may not apply to obesity in human patients. Also, our experimental scheme was not designed to address the effect of WW supplementation on lean mice; whether the observed changes in the gut microbiome and metabolite concentrations are irrespective of mice phenotype may warrant further research. The clinical relevance of this present work remains to be determined.

Future directions

In humans, obesity is a multifactorial disease that can be partly controlled with dietary modifications. This paper adds valuable information to the current literature with regard to the potential influence of WW consumption on the gut microbiota of obese diabetic mice. However, research is needed to investigate the effect of WW on obese human individuals.

The authors would like to express our deepest gratitude to the QIIME, Mothur and PICRUSt Developers and Help Forums for the creation of such exceptional computational tools and all the support provided. The authors would also like to thank Indira Mohanty and Alejandra Mencia for their technical assistance in the analysis of SCFAs and qPCR.

Additional Information and Declarations

Competing Interests

Author Contributions

Animal Ethics

DNA Deposition

The authors declare that they have no competing interests.

Jose F. Garcia-Mazcorro analyzed the data, wrote the paper, prepared figures and/or tables, reviewed drafts of the paper.

Ivan Ivanov analyzed the data, wrote the paper, reviewed drafts of the paper.

David A. Mills performed the experiments, analyzed the data, contributed reagents/materials/analysis tools, wrote the paper, reviewed drafts of the paper.

Giuliana Noratto conceived and designed the experiments, performed the experiments, analyzed the data, contributed reagents/materials/analysis tools, wrote the paper, reviewed drafts of the paper.

The following information was supplied relating to ethical approvals (i.e., approving body and any reference numbers):

The Institutional Animal Care Use Committee from Washington State University approved all experimental procedures (animal protocol approval number: 04436–001).

The following information was supplied regarding the deposition of DNA sequences:

NCBI SRA: PRJNA281761.

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
