# Peer review of "Influence of whole-wheat consumption on fecal microbial community structure of obese diabetic mice"

_PeerJ, doi:10.7717/peerj.1702_

## Round 0.1 · original submission · Minor Revisions

I agreed with all the reviewers comments and appropriate responses to these comments reflected in the manuscript should strengthen the findings and impact of this effort. For this reason, I have decided to ask for minor revisions .

Please pay particular attention to addressing these comments. In particular, please address how the findings can be interpreted based on Reviewer #1's comment about a missing control group for the lean mice with WW diet and this review also had strong comments which I agree with about the figures.

I would also suggest that the authors be very cautious of overstatements in the manuscript, mainly dealing with their findings of correlations vs. causations within the scope of the experimental design (line 332-333, 349-350 are specific examples). A critical examination of the discussion section I believe is necessary. This is in line with comments from reviewers (e.g., cage effects, missing controls) about being transparent of the limitations of this study and being clear about what this study shows.

·

Basic reporting

The results section will benefit from some minor reorganization (see comments below).

Experimental design

Good: Well written and organized, reproducible, clear hypothesis. Methods appropriate for the task, and of high standard.

Bad: The experiment lacks a key control group, mice with lean genotype fed with a whole wheat diet. Maybe whole wheat changes the community to the reported obese+ww stage irrespective of the mouse genotype?. or effects are only found with obese mice. The lack of the control makes it impossible to estimate the contribution of each factor, and restricts the limitations.

My suggestion is to either grow the mice under the needed conditions, or change the conclusions to reflect this limitation. The authors could probably cite other similar studies to fill this gap.

Validity of the findings

Statistical analysis is clear and appropriate. Transformation of data could be applied to reduce variability among treatments and use parametric test.

Data is publicly available, and methods and programs used are reproducible.

Additional comments

Abstract.
The authors use acronyms that are there used for the first time without defining them. E.g. WW for whole wheat, SCFA for short chain fatty acids. Since it is the abstract they can just remove the acronyms.
Line 39: replace “bacterial diversity” with “bacterial richness”.
Line 47: Replace “more relative abundances” with “higher abundances”

Introduction
Line 83: Use “changes” instead of “differences” since only changes are irreversible.
Methods
Line 114. Superscript is needed for the "db" in “Leprdb”.
Lines 191-192: Need to specify what correction was used in any case.

Results:
Fecal transformation composition
This paragraph need better organization since the results are a key part of the paper. The problems I see is that the authors repeat the results already presented in the figure give too much space for non-significant results and fail to explain sometimes where the differences among the three treatments occur. I recommend reorganizing figure 1 (see my comments below) to avoid repeating the results in both the text and figure, and in that way the statistics results could be moved to the figure to make the text more fluid. If the results are non-significant but important they should be reported but I don’t think it is appropriate to say that A was higher than B. The replication is pretty high (10 per sample) so I don’t think it is appropriate to stretch the results. Also, there are some transformations that could be applied to the abundances to normalize the variation and make a parametric test possible.

Figures
Fig1. Panels do not have letters (A or B). For panel A, I recommend a change to a clustered column type graph, that way the phyla can be more easily compared among treatments, otherwise the Firmicutes take too much space. With this type of graph the authors could also specify if the phyla abundances are significantly different using letters. Panel B has too many categories, and it is impossible to interpret because there are too many colors. A clustered column graph could help here too.

Fig3. It is not clear what is the message of this figure. The clustering results are also and better represented in the ordination, and there are many OTUs that show no significant difference among treatments. To make it clearer, you can probably remove the OTUs that do not change among treatments and just show those that do change. The clustering among samples could still be kept (you can pass the clustering based on all samples as an object to the heatmap function). Legend. More important than the program used for the visualization is the distance used. In heatmap.2 the default is Euclidean distance, since the authors were using Bray-Curtis they should use that one here too.

Fig4. The original purpose of rarefaction curves was to compare samples with different sample depth, but people use them indiscriminately to show the sampling performance. In such way, rarefaction curves are only useful when they reach saturation. Since no effort is made to estimate coverage, and this graph shows no saturation, little can be learnt from it. The authors could easily say on the text that there was no saturation and remove the figure.
Fig5. This plot is hard to visualize. 1. Red and orange colors are too similar to each other. 2. 3D plots are impossible to interpret without projections from the points to a plane. Since the third axis shows little variation it could be removed. 3. I would recommend a 2D plot with light background instead.

Fig6. Since all the figures have the same X and Y labels, you can probably use one on each side.

Tables:
Table 2. Footnote is unnecessary.
Table 3. Need post hoc tests after ANOVA to show where the differences are located. Also this is a really big table, I think it will be easier to read if the rows are ordered within levels two and three too.
Table 4. The authors could make it easier if they used letter codes to show the significant differences.

Discussion
The authors contrast the results from weighted and unweighted unifrac distances and focus on the differences in ordination to say that obesity changes abundances and diet changes richness. But they ignore that the variation explained by the axis changes dramatically between the two comparisons. Samples are separated in the first axis by obesity with 45% of the variation explained but when unweighted unifrac analysis is used the explanatory power drops to 11% so the changes are small.
SFCA analysis. Since the design does not include a lean group with whole wheat diet, it is impossible to know if the difference are due to obesity or diet.

The caveats section fits better before the summary.

Acknowledgments. Should not be used to acknowledge funder according to instructions for authors.

·

Basic reporting

No Comments

Experimental design

No Comments

Validity of the findings

No Comments

Additional comments

In the present manuscript “Influence of whole-wheat consumption on fecal microbial ecology of obese diabetic mice” Garcia-Mazcorro et al investigated the influence of whole-wheat (WW) consumption on the microbiota in obese mice. Lean mice served as a control. 16S rRNA gene-sequencing combined with qPCR analysis of specific groups of interest were applied to assess the bacterial community structure of fecal matter. Consumption of WW did not influence body weight nor stimulate a community shift towards lean communities. Both obese groups (WW and obese controls) were characterized by more similar communities (based on weighted unifrac distances and PICRUSt analysis) compared to those of lean mice. SCFA concentrations differ accordingly. Unweighted UniFracs, however, suggest unique communities in the WW group.

General comments: The manuscript is well structured and well written, applied methods and statistics are sound and the results are relevant for the scientific community. A few comments below:

- The cecum in mice is greatly enlarged harboring the majority of metabolic active bacteria. Assumptions on composition/activity of overall gut communities might not be well reflected in fecal matter. Especially metabolic measurements such as SCFA analysis are, hence, limited due to their fast turn over rates; it is less of an issue for DNA based analysis (16S/qPCR). This should be mentioned somewhere in the manuscript.

- Did the authors observe any cage effects (especially the obese controls cluster in 2 separate groups – Figure 3 & 5)? It seems that groups show differences in dispersion where the WW group clusters the most concise – this might be an interesting result and I encourage the authors to add dispersion analysis (if not an artifact from cage effects).

- It is appreciated that authors do not shy away from showing both UniFrac distances as combined they indeed provide more comprehensive insights on community structure. The WW group is distinct to other groups in the unweighted format – are there any low abundant taxa unique to this group? Conventional statistics often fail to reveal low abundant taxa, but they might be important here so e.g. a simple Venn diagram for taxa presence between groups might be a valuable addition?

Minor comments:

- The axis scale in Figure 6 is misleading as they suggest similar abundances of all taxa investigated.

- I would omit “microbial ecology” from the title and change it to e.g. “microbial community structure” – microbial ecology is a whole research field and only a tiny part is considered here. Also, I would not mention the total number of OTUs in the abstract as a considerable amount is probably artifact due to sequencing errors (even global singletons are considered here).

- Some language/expression adjustments are encouraged:
. l 77 –omit “environmental” from pathogen
. l 308 – change “microbial ecology” to bacterial community structure
. l 325-327 – restructure the sentence not clear at the moment
. l 344/345 – “communities by what can live in them” sounds weird.
. 347/348 – I thought bacterial richness (number OTUs) is not affected? – rephrase.
. 358 why “in turn”?

---

## Round 0.2 · Minor Revisions

Please address comments on Figure 4 & 6. Otherwise, this manuscript looks good for acceptance.

·

Basic reporting

No comments

Experimental design

No comments

Validity of the findings

No comments

Additional comments

The authors did a good job in responding to my comments and in modifying the text, figures, and tables to make the paper easier to understand.

·

Basic reporting

see below

Experimental design

see below

Validity of the findings

see below

Additional comments

Most of my comments were addressed so I suggest to accept this manuscript for publication. However, I still have two suggestions for Figure 4 & 6:
Figure 4: in my opinion this Figure can be omitted or reduced to the box plots as the message of the curves is not clear (is the text "Rarefied plots of number of OTUs showed that more than the 3000 sequences per sample used in this study are needed to fully describe the fecal microbiota of all mice" referring to that?). Also, in the text it should say "dispersion in number of OTUs detected". I would also omit the definition of 97% similarity for species as this is highly debated and doubtful whether a clear species cut-off exists.
Axis in Figure 6: I still do not understand the labelling - for me it reads like this: in lean mice 95% belong to Bacteroidetes, 95% to Firmicutes, 87% to Ruminococcaceae, 85% to B. fragilis, 140% to Akkermansia, ..., which is obviously wrong.

---

## Round 0.3 · accepted · Accept

Thank you for your resubmission. The reviewers and I are happy to accept this manuscript.